# Paternal adverse childhood experiences and offspring's attentional disengagement from faces at 8 months—Results from the FinnBrain Birth Cohort Study

Magdalena Klimek[1,2,3]*, Hasse Karlsson[2,3,4,5], Linnea Karlsson[2,3,6,7], Riikka Korja[2,8,9], Saara Nolvi[2,3,8,9], Tuomo Häikiö[8], Jetro J. Tuulari[2,3,4,9,10,11], Eeva-Leena Kataja[2,3]

1 Department of Environmental Health, Faculty of Health Sciences, Institute of Public Health, Jagiellonian University Medical College, Krakow, Poland, 2 FinnBrain Birth Cohort Study, Turku Brain and Mind Center, Department of Clinical Medicine, University of Turku and Turku University Hospital, Turku, Finland, 3 Centre for Population Health Research, Turku University Hospital and University of Turku, Turku, Finland, 4 Department of Clinical Medicine, University of Turku, and Turku University Hospital, Turku, Finland, 5 Department of Psychiatry, Turku University Hospital & University of Turku, Turku, Finland, 6 Department of Public Health, University of Turku and Turku University Hospital, Turku, Finland, 7 Department of Child Psychiatry, Turku University Hospital and University of Turku, Turku, Finland, 8 Department of Psychology and Speech-language Pathology, University of Turku, Turku, Finland, 9 The Centre of Excellence for Learning Dynamics and Intervention Research (InterLearn), University of Turku and University of Jyväskylä, Finland, 10 Clinical Neurosciences, University of Turku, Turku, Finland, 11 Neurocenter, Turku University Hospital, Turku, Finland

* magda.klimek@uj.edu.pl

## Abstract

Paternal adverse childhood experiences (ACEs) have been recently linked to offspring's brain development. Yet, none of the previous studies in humans have explored the association between paternal ACEs and a child's attentional bias for facial expressions of emotion. Our study fills this gap. Data were collected from 239 fathers (mean age 32.15; SD 5.04) and their children at 8 months of age who were part of the FinnBrain Birth Cohort Study. Paternal ACEs were evaluated using the Trauma and Distress Scale (TADS) in five domains: emotional and physical neglect, emotional and physical abuse, and sexual abuse. In children, eye-tracking was used to study attentional engagement to emotional faces vs. non-faces and distractors, and to calculate face and fear bias indices. Hierarchical linear regression and the Mann-Whitney U test were used for analyses. A negative association between paternal sexual abuse and face bias was found in children (p = 0.043), when paternal postpartum anxiety and sex of the child were controlled, however the effect size was rather low. Additionally, daughters (n = 6) of sexually abused fathers expressed lower face bias (p = 0.02) and higher fear bias (p = 0.04) than daughters of sexually non-abused fathers. Our preliminary exploration suggests a potential intergenerational effect of paternal exposure to sexual abuse on the processing of facial expression

**Data availability statement:** The Finnish law and ethical permissions do not allow open sharing of the data used in this study, but data access is possible via formal material transfer agreements (MTA). Investigators that wish to access the data are encouraged to contact FinnBrain board through Eija Jossandt (eianha@utu.fi).

**Funding:** This study was supported by the Academy of Finland in the form of a grant awarded to LK and E-LK #308589, Strategic Research Council (SRC) within the Research Council of Finland awarded to LK (#352648, #352655), Research Council of Finland awarded to RK (#346121, #308252), the Signe and Ane Gyllenberg Foundation (LK,E-LK,RK,JJT,SN), Finnish State Grants for Clinical Research (VTR) (LK,E-LK,RK,JJT,SN), Finnish Cultural Foundation (RK,SN), Yrjö Jahnsson Foundation (LK,RK,SN), Emil Aaltonen Foundation (JJT,SN). The study was also supported by the Jagiellonian University International Relations Office within the Erasmus+ program awarded to MK, the "Better research for better quality of life – qLIFE" awarded to MK (qL.1.4.2024.7), and Polish Ministry of Science and Higher Education grant awarded to MK (N43/DBS/000303). The specific roles of the authors are articulated in the 'author contributions' section. The funders had no role in study design, data collection and analysis, decision to publish, or preparation of the manuscript.

**Competing interests:** The authors have declared that no competing interests exist.

among daughters at the age of 8 months, yet the results require further confirmatory analyses, especially in a larger study group of ACEs-exposed individuals.

## Introduction

Accumulating research presents well-established evidence that individuals exposed to adverse childhood experiences (ACEs) are at higher risk of having mental health disorders, e.g., depression, anxiety, post-traumatic stress disorder or personality disorders (for a review see: [1,2]). Key forms of ACEs are formed by childhood maltreatment exposure that entails emotional, physical, or sexual abuse, as well as emotional or physical neglect [3]. It is hypothesized that such risks could result from the fact that a stressful environment during childhood might affect the cognitive and social development of a child, involving the perception and processing of emotions.

Attentional processing of emotional signals from faces is one key cognitive trait emerging early in human development and influencing later social and emotional development [4]. The key components of the face and emotion detection networks begin to emerge early in development. Infants show preference for faces soon after birth, and this cognitive adaptation is likely developed to facilitate engagement in vital interactions with the caregivers [5]. In addition to enhancing contact between the infant and the caregiver, an important function of the face detection systems is to scan incoming sensory information in order to detect the presence of biologically relevant features (e.g., stimuli that represent a threat to wellbeing) and grant them priority in access to attention and awareness [6]. Hence, rapid detection of fear or threat is considered a vital attentional function, enhancing survival [7]. After the second half of the first year infants start to disengage their attention less probably or more slowly from fearful vs. non-fearful faces towards salient distracting stimuli [8]. Although these biases for faces and fear or threat are normative to some extent, environmental factors such as maternal mental health seem to shape individual variation already in infancy [9,10]. A recent systematic review revealed that children exposed to maltreatment present a preferential attention and increased sensitivity towards negative facial expressions, principally fear and anger, potentially as a result of experience-specific information-processing bias [11].

It has been suggested that the effects of adverse experiences during childhood could be transmitted across generations via developmental programming processes occurring in utero [12–14]. Most of the studies exploring the intergenerational transmission of ACEs, however, are embedded within mother-child dyads, showing that maternal exposure to childhood adversities is related to offspring's behavioral problems [15], emotional reactivity [16], risk of psychopathology [17] as well as deviances in brain structure and function (for a review see: [14]). To our knowledge, not much is known about the paternal influences in shaping infant face and fear processing. A recently emerged paradigm of POHaD (Paternal Origins of Health and Disease) points attention to the fact that early developmental programming is not

fully dependent and limited only to maternal characteristics [18]. It is also influenced by paternal factors which have to be investigated in the coming years as the research in this area is scarce [18,19]. Several pathways that might stand behind the paternal effects on the offspring phenotype are suggested, including social, historical, as well as biological interplay [19]. Paternal lifetime exposures have also been studied in an intergenerational manner, namely the effects of paternal characteristics on the offspring's phenotype [20]. Paternal exposure to adversities during early years has been linked to, e.g., offspring's anxiety and depression (for a review see: [21]), neuropsychiatric disorders, and developmental delays [22]. This sheds light on the fact that paternal exposure to ACEs could also shape an offspring's socio-emotional and cognitive development, while to date, not many studies have linked paternal ACEs with their child's early development. Our recent findings in newborns aged 2–5 weeks who underwent a magnetic resonance imaging procedure showed that higher paternal ACEs were associated with faster white matter maturation in the corpus callosum [23], larger grey matter volume [24] as well as higher left amygdala volumes [25]. How these changes influence child development in a long-term manner is of interest for understanding the mechanisms of paternal transmission of ACEs. Our study from the same cohort as in this study indicated that individual variation already in newborns' amygdala volume is associated with how infants disengaged their attention from fearful facial expressions towards salient distractors while no associations were detected for happy and neutral faces [26]. Hence, infant face and fear processing might be early intermediate phenotypes indicating risk for later social emotional problems, while to the best of our knowledge, no previous studies have explored the association between paternal early-life adversities and emotional processing in the offspring. We could, however, expect that paternal ACEs might program the early-developing neural face and threat detection system due to their vital role in enhancing survival in the offspring.

Going further, girls and boys might process and react to the visual stimuli differently in infancy as a result of fetal programming effects [27] or parental pre- and postnatal mental health [28]. Hence, girls might become more reactive to social signals and fear and present more alert attention, contrary to boys, who are less responsive to fear [29]. Additionally, if paternal transgenerational inheritance is embedded in epigenetic processes, those processes were also suggested to act sex-specifically [30,31]. For example, animal models have provided evidence of sexually differentiated levels of DNA methylation in the brain between females and males [32]. Additionally, the epigenetic regulation platform could potentially act via sex hormones, which control sexually dimorphic development and functioning of the brain [33]. Evidence in humans, however, is still scarce and more studies are needed to investigate the role of epigenetic mechanisms in sex-dependent brain development.

Our current study aims to fill the gap in the literature by investigating the relationship between cumulative paternal ACEs and attentional processing of emotion signals from faces in their offspring, namely attentional engagement to emotional facial expressions, including fear. Furthermore, we explore the potential sex difference in response to paternal ACEs. We examined 8-month-old infants, as around this age, infants start to differentiate between facial expressions and emotions as well as disengage their attention from faces towards distractors, an effect influenced by the presented expression [8].

## Methods

### Participants

Parents at gestational week (gwk) 12 and their later-born children were recruited between 07.12.2011–08.06.2015 within the FinnBrain Birth Cohort Study (https://sites.utu.fi/finnbrain/en/), a population-based pregnancy cohort established in Finland, consisting of White, ethnically European individuals (the majority of whom were of Finnish origin, 2.5% of a sample were of other nationalities, e.g., Swedish, Lithuanian, Russian) [34]. All eligible participants were contacted in the area of the city of Turku and Åland Islands region (Southwest Finland). The participants were recruited by research nurses in health clinics. No siblings were involved in the study, besides one pair of twins. The sample in the current study originally included 2623 fathers. For 1953 fathers questionnaire data on their ACEs was available. Out of this, we had eye-tracking

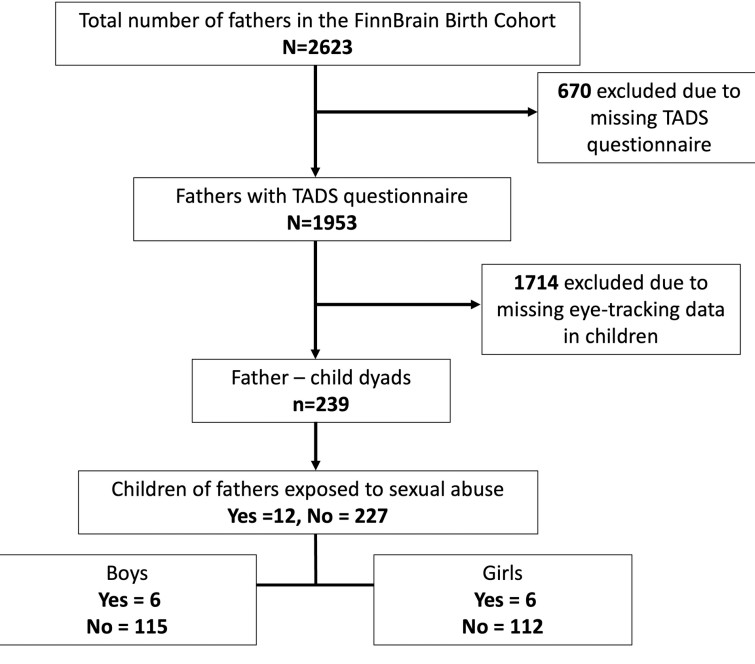

data for 239 of their infants who participated in a Child Development and Parental Functioning study visit at 8 months. The sex of the child was collected from birth records obtained from the Finnish Medical Birth Register. The mean paternal age was 32.15 (SD 5.04) and 51% of the infants were boys. The prevalence of premature birth (before gwk 37) was relatively low and was observed in 3.8% of the children. A flowchart presenting the formulation of a final study sample (after several exclusions due to missing data) is presented in Fig 1. This study was approved by the Ethics Committee of the Hospital District of Southwest Finland (ETMK: 57/180/2011 for the questionnaire data and ETMK: 107/180/2012 for the eye-tracking data). All research was performed following the Declaration of Helsinki and written informed consent was obtained from each adult participant. Also, a written, informed consent was obtained from the parents and on behalf of their child.

## Paternal adverse childhood experiences, ACEs

Paternal ACEs were evaluated at gwk 14 using the Trauma and Distress Scale (TADS) in five core domains: emotional and physical neglect, emotional and physical abuse, and sexual abuse [35]. Cumulative exposure to ACEs was calculated based on individual factor scores for three age ranges for the timing the adverse childhood experiences potentially occurred (0–6 years, 7–12 years, and 13–18 years) – such an approach is an extension of our previous analysis based on those age ranges [25]. Fathers scoring zero points in the TADS questionnaire (either in general score or in five core domains separately) were considered as "non-abused during childhood", while fathers scoring at least one point were studied as "abused during childhood". TADS total score, as well as separate scores for five core domains, were used in the analyses (Table 1). Such an approach was justified as the Pearson's correlation coefficient between the core domains ranged from 0.116 to 0.686, meaning a rather small to moderate overlap between those subcategories. This might be useful in unveiling the separate effects of selected stressors, which otherwise might have been blurred in a larger context of "general childhood adversities" used in previous studies (for a review see: [36]). The distributions of TADS questionnaire core domains are presented in S1-S5 Figs in the S1 File.

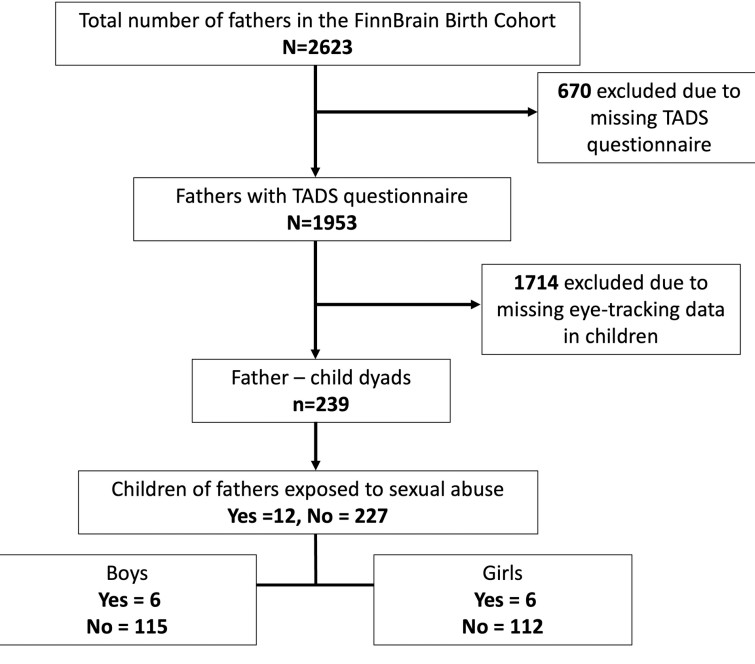

**Fig 1. A flowchart presenting the formulation of a final study sample.**

**Table 1. Characteristic of the study sample.**

| | Total sample (N = 239) | | | | | |
|---|---|---|---|---|---|---|
| Child sex (%, boys) | 50.63% | | | | | |
| Maternal education level – mid and low (N, %) | 71 (29.7%) | | | | | |
| Maternal education level – vocational and high (N, %) | 167 (69.9%) | | | | | |
| Paternal education level – mid and low (N, %) | 105 (43.9%) | | | | | |
| Paternal education level – mid and low (N, %) | 133 (55.7%) | | | | | |
| | N | Mean | SD | Min. | Max. | S.E. |
| Paternal age | 239 | 32.15 | 5.04 | 17 | 48 | 0.33 |
| Estimated paternal income* | 239 | 2.05 | 0.73 | 1 | 4 | 0.16 |
| Paternal depression (EPDS), 6 months postpartum (range: 0–30) | 146 | 3.57 | 3.68 | 0 | 18 | 0.30 |
| Paternal anxiety (SCL-90), 6 months postpartum (range: 0–40) | 146 | 2.28 | 3.44 | 0 | 24 | 0.28 |
| Paternal TADS, total score (range: 0–172) | 239 | 21.00 | 13.21 | 1 | 71 | 0.86 |
| Paternal emotional abuse (range: 0–20) | 239 | 1.69 | 2.40 | 0 | 11 | 0.16 |
| Paternal emotional neglect (range: 0–20) | 239 | 3.83 | 3.50 | 0 | 15 | 0.22 |
| Paternal physical abuse (range: 0–20) | 239 | 1.42 | 2.07 | 0 | 13 | 0.13 |
| Paternal physical neglect (range: 0–20) | 239 | 2.72 | 2.18 | 0 | 9 | 0.14 |
| Paternal sexual abuse (range: 0–20) | 239 | 0.13 | 0.69 | 0 | 8 | 0.04 |

* Income categories (monthly paternal income): 1 = 1500 EUR or less, 2 = 1501−2500 EUR, 3 = 2501−3500 EUR, 4 = more than 3500 EUR; EPDS – Edinburgh Postnatal Depression Scale; SCL-90 – Symptom Checklist 90; TADS – Trauma and Distress Scale; S.E. - standard error.

## Eye-tracking assessment

Attentional engagement to emotional facial expressions was measured using an eye-tracking procedure at child's 8 months of age. Children were positioned at their parent's lap (the majority of whom were mothers, very rarely fathers or grandmothers) at a distance of 50–70 cm from the eye tracker (EyeLink 1000 +, SR Research Ltd., Toronto, ON, Canada), and a sampling frequency of 500 Hz was used. A five-point calibration procedure was used before the start of the experiment to ensure the quality of the measurement. An audiovisual animation (i.e., a dog or a duck) was presented sequentially in five different spots on the screen. When necessary, the calibration was replicated before actual testing and also during the measurement process. Small breaks were allowed during the measurement procedure if needed. The eye-tracking laboratory was dimly lit and the researcher was positioned by an independent host computer next to the parent and the child, but was separated by a curtain in order to avoid any disruption. The eye-tracking data collection lasted between 11.04.2013 and 30.06.2016.

An emotional overlap paradigm [37] was used to assess infant attention disengagement from a centrally presented face or a scrambled face control stimulus (CS) to a lateral distractor (See Fig 2). A set of photographs, presenting two different females, with faces of different emotions (happy, HA; fearful FE; neutral, NE) and phase-scrambled control pictures, was used. Altogether, a set of 48 trials was presented, 12 trials per condition (each emotion and the control picture), comprising 18 photographs of each woman, and 12 scrambled face control pictures, in a semi-random order.

The eye-tracking data was pre-processed as follows: First, the trial data included the xy coordinates of the participants' trial as well as timestamps for the onset times of central and lateral pictures, which represent the appearance of the face or nonface on the screen, and extended to the end of the analysis period, that is, gaze disengagement from the central to lateral stimulus or if a gaze disengagement was not observed, 1,000 ms after the lateral distractor appeared. Second, trials had to have a sufficient number of valid samples in the gaze data with no gaps greater than 200 ms. This meant that gaps in the data were extrapolated by the analysis script by carrying the last recorded sample forward, and that the trial was flagged as invalid and excluded from subsequent analyses if the gap was greater than 200 ms. Third, if

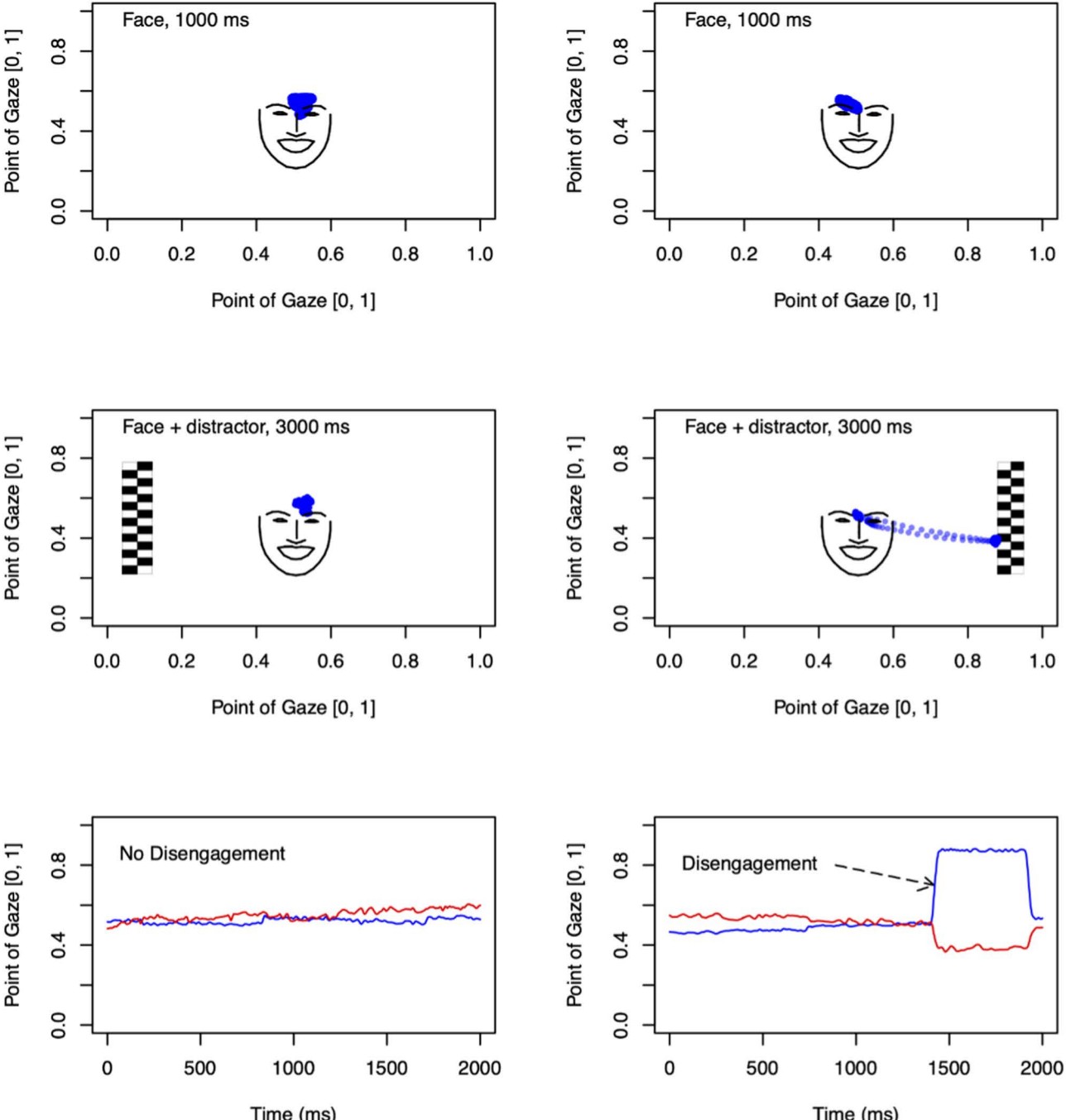

**Fig 2. An illustration of the eye-tracking procedure [38].** Participants were presented with a face or a non-face in the center of the screen. One second later, a "distractor" was added to the left or to the right. Left: An example of a "no disengagement" trial in which the gaze does not shift from the central to the lateral stimulus. Right: An example of a "disengagement" trial in which the gaze shifts rapidly from the central to the lateral stimulus. In the two top rows, the blue dots denote the gaze location over the trial period. In the bottom row, the x and y coordinates of the point of gaze on the display are shown by the blue and red lines, respectively. The outline and key landmarks of the face stimulus are shown based on values extracted by the Open-Face 2.2.0 toolkit [39].

disengagement occurred during the trial, to ensure that the measurement was included in the analysis, the exact timing of the eye movement from the central to the lateral stimulus was required to be known for the trial. That is, the trial was rejected if the eye movement occurred during a period of missing or extrapolated gaze data.

Eye-tracking variables: Prior studies based on the current methodology, assessed infants' attentional disengagement by processing the trial-level eye-tracking data and coding the data into a binary disengagement value (0/1). This was determined on whether or not the gaze shifted from the central to the lateral stimulus (i.e., disengagement probability, DP) or by extracting the latency of the gaze shift. In the current study, as our primary measures of disengagement, we included binary disengagement variables and mean DPs. Further, following Yrttiaho and colleagues [40], we calculated "Face bias" and "Fear bias" indices as follows: the face bias was defined as the mean difference in the DPs in the control non-face condition vs. (non-fearful) face condition (i.e., p[saccade/CS] – p[saccade/NE&HA]) and the fear bias was defined as the mean difference in the DPs in the (non-fearful) face condition vs. fearful face condition (i.e., p[saccade/NE&HA] – p[saccade/FE]). Only participants with a minimum of three successful trials in each experimental condition were included in the data.

## Covariates

Prior to the analyses, based on the existing literature (e.g., [23,41,42] several covariates were chosen and collected via questionnaire, including monthly paternal income (as a proxy of socioeconomic status), paternal and maternal age, paternal and maternal depression and anxiety at 6 months postpartum, and maternal childhood stress. Both in fathers and mothers, postpartum depression was evaluated using the Edinburgh Postnatal Depression Scale (EPDS) [43], anxiety levels were scored using the Symptom Checklist-90 (SCL-90) [44] and childhood adversity was assed using the Trauma and Distress Scale (TADS) [35]. For the sake of representativeness, we have compared the studied group (n = 239) with the original total sample of fathers recruited within the FinnBrain Birth Cohort Study (N = 2623). There was no difference between the studied group (n = 239) and the total sample (N = 2623) in paternal age at birth (p = 0.56), estimated paternal income (p = 0.31), paternal depression (p = 0.69), and anxiety level (p = 0.37) at 6 months postpartum.

## Statistical analyses

IBM SPSS Statistics version 28.0.1.0 was used to perform the analyses. Spearman rank correlation analysis, hierarchical linear regression models, and the Mann-Whitney U test were performed. The Spearman rank correlation and Mann-Whitney U test were chosen for the sake of the paternal TADS scores skewness (see: S1-S5 Figs in S1 File). First, Spearman rank correlation analysis between paternal TADS sum score, five core subdomains, and parameters of emotion processing is presented in Table 2. In the next step, the correlations between selected covariates involving paternal age, depression, and anxiety at 6 months postpartum, as well as maternal age, childhood adversity, depression, and anxiety at 6 months postpartum and child's attentional disengagement from faces measures were explored. If a statistically significant correlation was observed, then the covariate was included in the final statistical model. The sex of the child was included in all analysed models. Next, the main effects of paternal TADS (total score or core domains scores: emotional abuse, emotional neglect, physical abuse, and physical neglect) on child's attentional disengagement from faces measures were tested in hierarchical linear regression models. Linear analyses were preceded by the evaluation of normality of distribution, equality of variances (Levene's test), and collinearity diagnosis (Variance Inflation Factor, VIF). Due to high collinearity (VIF score >10), the TADS total score and core domains subscores were entered in separate models (TADS total score as Model 1, and five core subdomain scores as Model 2) (Table 3). Five core subdomains were entered into the models together as the VIF factor did not reach the level of >3 for each subdomain, and also the Spearman rank correlation coefficients between them were not very strong (Table 2). Next, if a statistically significant correlation was observed between the covariate and dependent variable, the covariate was entered, and in the second step, paternal

**Table 2. Spearman rank correlation matrix of the study variables (N = 239).**

| Variables | 1 | 2 | 3 | 4 | 5 | 6 | 7 | 8 | 9 | 10 | 11 |
|---|---|---|---|---|---|---|---|---|---|---|---|
| **1 TADS sum score** | 1.00 | | | | | | | | | | |
| **2 Emotional neglect** | .84** | 1.00 | | | | | | | | | |
| **3 Emotional abuse** | .72** | .55** | 1.00 | | | | | | | | |
| **4 Physical neglect** | .73** | .64** | .43** | 1.00 | | | | | | | |
| **5 Physical abuse** | .57** | .33** | .54** | .36** | 1.00 | | | | | | |
| **6 Sexual abuse** | .18** | .05 | .12 | .13 | .18** | 1.00 | | | | | |
| **7 DP Neutral** | .05 | .02 | −.005 | .003 | −.007 | .15* | 1.00 | | | | |
| **8 DP Happy** | .11 | .06 | .11 | .06 | .18** | .11 | .72** | 1.00 | | | |
| **9 DP Fearful** | .09 | .02 | .05 | .07 | .10 | .05 | .70** | .62** | 1.00 | | |
| **10 Face bias** | .03 | .10 | −.04 | −.14* | −.14* | −.14* | −.58** | −.60** | −.44** | 1.00 | |
| **11 Fear bias** | −.02 | .03 | −.01 | −.001 | −.001 | .10 | .10 | .20** | −.55** | −.09 | 1.00 |

*p < 0.05 (2-tailed), **p < 0.01 (2-tailed).

DP – disengagement probability.

TADS – Trauma and Distress Scale.

TADS score (total score or core domains scores) was added. Last, the Mann-Whitney U test was performed to evaluate the differences in attentional disengagement measures between boys and girls. The internal consistency of all questionnaires used in this study was evaluated by Cronbach's alpha and the obtained values were as follows: maternal EPDS α = 0.88 and SCL-90 α = 0.87 at six months postpartum, paternal EPDS α = 0.83 and SCL-90 α = 0.85 at six months postpartum, maternal TADS α = 0.92 and paternal TADS α = 0.89.

## Results

Characteristics of the study sample are presented in Table 1. Paternal monthly income, paternal age, and paternal postpartum depression were not significantly correlated with any attentional disengagement measures nor face or fear bias (rs < 0.14; ps > 0.10). Only a significant negative association between paternal postpartum anxiety and infant face bias was detected (r = −0.264; p = 0.001; 95%CI −0.409 to −0.106), therefore paternal anxiety was included in the further analysis of face bias as a covariate. Maternal age, childhood adversity, and postpartum anxiety were not related to any parameters of emotion processing in their offspring (p ranged from 0.18 to 0.99), except from postpartum depression at six months associated with higher fear bias (r = 0.166; p = 0.019; 95%CI 0.028 to 0.298) which was later included as a covariate in models involving fear bias.

Hierarchical linear regression models revealed that offspring's disengagement from neutral faces was predicted by paternal sexual abuse in childhood (standardized ß = 0.137; p = 0.045), taking into account other TADS subscores, that is, emotional and physical neglect, as well as emotional and physical abuse and sex of the child. Similarly, offspring's face bias was predicted by paternal childhood sexual abuse score (standardized ß = −0.164; p = 0.043) when controlling for paternal postpartum anxiety, sex of the child and other TADS subscores. No other statistically significant associations were detected (Table 3).

When sons and daughters of sexually abused and non-abused fathers were examined separately, we found that daughters of sexually abused fathers had higher disengagement probability than daughters of sexually non-abused fathers for neutral faces U = 145.50; p = 0.02), as well as lower face bias (U = 151.50; p = 0.02) and higher fear bias (U = 171.50; p = 0.04) (Table 4). No significant results were observed in sons of fathers exposed or non-exposed to sexual abuse during childhood (Table 5).

**Table 3. Hierarchical linear regression of paternal adverse childhood experiences and eye-tracking of attention to emotional faces in 8-month-old children (N = 239).**

| | $R^2$ | $\Delta R^2$ | SE | Unstandardized β | Standardized β | t | p | F for change in $R^2$ |
|---|---|---|---|---|---|---|---|---|
| **DP Neutral** | | | | | | | | |
| *Model 1:* DP Neutral ~ Paternal TADS + sex of the child | | | | | | | | |
| Paternal TADS, total score | | | 0.06 | 0.0001 | 0.010 | 0.15 | 0.88 | |
| Sex of the child | | | 0.03 | −0.04 | −0.07 | −1.10 | 0.27 | |
| *Model 2:* DP Neutral ~ paternal emotional abuse + paternal emotional neglect + paternal physical abuse + paternal physical neglect + paternal sexual abuse + sex of the child | | | | | | | | |
| Paternal emotional abuse | | | 0.01 | −0.009 | −0.084 | −0.83 | 0.408 | |
| Paternal emotional neglect | | | 0.01 | 0.002 | 0.023 | 0.22 | 0.825 | |
| Paternal physical abuse | | | 0.01 | 0.001 | −0.007 | −0.08 | 0.939 | |
| Paternal physical neglect | | | 0.01 | 0.001 | 0.008 | 0.09 | 0.926 | |
| Paternal sexual abuse | | | 0.03 | 0.053 | 0.137 | 2.02 | **0.045** | |
| Sex of the child | | | 0.03 | −0.039 | −0.074 | −1.12 | 0.264 | |
| **DP Happy** | | | | | | | | |
| *Model 1:* DP Happy ~ Paternal TADS + sex of the child | | | | | | | | |
| Paternal TADS, total score | | | 0.001 | 0.002 | 0.087 | 1.34 | 0.182 | |
| Sex of the child | | | 0.33 | −0.019 | −0.038 | −0.58 | 0.564 | |
| *Model 2:* DP Happy ~ paternal emotional abuse + paternal emotional neglect + paternal physical abuse + paternal physical neglect + paternal sexual abuse + sex of the child | | | | | | | | |
| Paternal emotional abuse | | | 0.01 | 0.004 | 0.042 | 0.42 | 0.676 | |
| Paternal emotional neglect | | | 0.01 | 0.0001 | −0.002 | −0.02 | 0.983 | |
| Paternal physical abuse | | | 0.01 | 0.005 | 0.038 | 0.44 | 0.659 | |
| Paternal physical neglect | | | 0.01 | −0.002 | −0.017 | −0.18 | 0.853 | |
| Paternal sexual abuse | | | 0.02 | 0.040 | 0.109 | 1.60 | 0.112 | |
| Sex of the child | | | 0.03 | −0.020 | −0.039 | −0.59 | 0.557 | |
| **DP Fearful** | | | | | | | | |
| Model 1: DP Fearful ~ Paternal TADS + sex of the child | | | | | | | | |
| Paternal TADS, total score | | | 0.001 | 0.001 | 0.034 | 0.52 | 0.604 | |
| Sex of the child | | | 0.04 | −0.011 | −0.019 | −0.30 | 0.766 | |
| Model 2: DP Fearful ~ paternal emotional abuse + paternal emotional neglect + paternal physical abuse + paternal physical neglect + paternal sexual abuse + sex of the child | | | | | | | | |
| Paternal emotional abuse | | | 0.01 | −0.001 | −0.005 | −0.05 | 0.959 | |
| Paternal emotional neglect | | | 0.009 | −0.001 | −0.015 | −0.14 | 0.891 | |
| Paternal physical abuse | | | 0.01 | −0.001 | −0.009 | −0.10 | 0.918 | |
| Paternal physical neglect | | | 0.01 | 0.006 | 0.043 | 0.48 | 0.633 | |
| Paternal sexual abuse | | | 0.03 | 0.006 | 0.015 | 0.22 | 0.824 | |
| Sex of the child | | | 0.04 | −0.009 | −0.017 | −0.25 | 0.804 | |
| **Face bias** | | | | | | | | |
| *Model 1:* Step 1 Face bias ~ Paternal anxiety + sex of the child | 0.067 | 0.080 | | | | | | 6.177* |
| Paternal anxiety (SCL-90) | | | 0.006 | −0.019 | −0.266 | −3.31 | **0.001** | |
| Sex of the child | | | 0.040 | 0.049 | 0.099 | 1.23 | 0.220 | |
| *Step 2* Face bias ~ Paternal anxiety + paternal TADS + sex of the child | 0.062 | 0.002 | | | | | | 0.326 |
| Paternal anxiety (SCL-90) | | | 0.006 | −0.020 | −0.276 | −3.35 | **0.001** | |
| Paternal TADS, total score | | | 0.002 | 0.001 | 0.047 | 0.57 | 0.569 | |
| Sex of the child | | | 0.040 | 0.048 | 0.097 | 1.21 | 0.230 | |

*(Continued)*

| | $R^2$ | $\Delta R^2$ | SE | Unstandardized β | Standardized β | t | p | F for change in $R^2$ |
|---|---|---|---|---|---|---|---|---|
| **Model 2:** Face bias ~ Paternal anxiety + paternal emotional abuse + paternal emotional neglect + paternal physical abuse + paternal physical neglect + paternal sexual abuse + sex of the child | | | | | | | | |
| Paternal anxiety (SCL-90) | | | 0.006 | −0.019 | −0.267 | −3.256 | **0.001** | |
| Paternal emotional abuse | | | 0.014 | −0.010 | −0.086 | −0.744 | 0.458 | |
| Paternal emotional neglect | | | 0.009 | 0.013 | 0.180 | 1.530 | 0.128 | |
| Paternal physical abuse | | | 0.013 | −0.001 | −0.004 | −0.044 | 0.965 | |
| Paternal physical neglect | | | 0.012 | −0.003 | −0.025 | −0.239 | 0.812 | |
| Paternal sexual abuse | | | 0.050 | −0.103 | −0.164 | −2.045 | **0.043** | |
| Sex of the child | | | 0.040 | 0.049 | 0.099 | 1.213 | 0.227 | |
| **Fear bias** | | | | | | | | |
| **Model 1:** Step 1 Fear bias ~ Maternal EPDS + sex of the child | 0.028 | 0.028 | | | | | | 2.782 |
| Maternal EPDS | | | 0.003 | 0.007 | 0.167 | | **0.020** | |
| Sex of the child | | | 0.028 | 0.002 | 0.004 | | 0.958 | |
| Step 2 Fear bias ~ Maternal EDPS + paternal TADS + sex of the child | 0.029 | 0.001 | | | | | | 0.208 |
| Maternal EPDS | | | 0.003 | 0.008 | 0.174 | 2.381 | **0.018** | |
| Paternal TADS, total score | | | 0.001 | −0.0001 | −0.033 | −0.456 | 0.649 | |
| Sex of the child | | | 0.028 | 0.002 | 0.006 | 0.079 | 0.937 | |
| **Model 2:** Fear bias ~ Maternal EPDS + paternal emotional abuse + paternal emotional neglect + paternal physical abuse + paternal physical neglect + paternal sexual abuse + sex of the child | | | | | | | | |
| Maternal EPDS | | | 0.003 | 0.007 | 0.149 | 2.004 | **0.046** | |
| Paternal emotional abuse | | | 0.009 | −0.006 | −0.069 | −0.069 | 0.536 | |
| Paternal emotional neglect | | | 0.006 | 0.003 | 0.048 | 0.419 | 0.676 | |
| Paternal physical abuse | | | 0.009 | 0.005 | 0.054 | 0.572 | 0.568 | |
| Paternal physical neglect | | | 0.009 | −0.008 | −0.092 | −0.918 | 0.360 | |
| Paternal sexual abuse | | | 0.020 | 0.032 | 0.120 | 1.594 | 0.112 | |
| Sex of the child | | | 0.029 | −0.001 | 0.149 | −0.021 | 0.983 | |

DP – disengagement probability; TADS – Trauma and Distress Scale; SCL-90 – Symptom Checklist 90 at 6 months post-partum, EPDS – Edinburgh Postnatal Depression Scale.

* p < 0.001.

**Table 4. Eye-tracking measures in daughters of sexually and non-sexually abused fathers.**

| | Daughters of sexually abused fathers n = 6 | | Daughters of sexually non-abused fathers n = 112 | | Mann-Whitney U | Z | p-value (2-tailed) |
|---|---|---|---|---|---|---|---|
| | **Mean Rank** | **Sum of Ranks** | **Mean Rank** | **Sum of Ranks** | | | |
| DP Neutral | 91.25 | 547.50 | 57.80 | 6473.50 | 145.50 | −2.34 | **0.02** |
| DP Happy | 84.50 | 507 | 58.16 | 6514 | 186 | −1.84 | 0.06 |
| DP Fearful | 69.83 | 419 | 58.95 | 6602 | 274 | −0.76 | 0.45 |
| Face bias | 28.75 | 172.50 | 61.15 | 6848.50 | 151.50 | −2.26 | **0.02** |
| Fear bias | 86.92 | 521.50 | 58.03 | 6499.50 | 171.50 | −2.02 | **0.04** |

DP – disengagement probability.

**Table 5. Eye-tracking measures in sons of sexually and non-sexually abused fathers.**

| | Sons of sexually abused fathers n = 6 | | Sons of sexually non-abused fathers n = 115 | | Mann-Whitney U | Z | p-value (2-tailed) |
|---|---|---|---|---|---|---|---|
| | Mean Rank | Sum of Ranks | Mean Rank | Sum of Ranks | | | |
| DP Neutral | 74.83 | 443 | 60.33 | 6938 | 268 | −0.92 | 0.36 |
| DP Happy | 64.25 | 385.50 | 60.83 | 6995.50 | 325.50 | −0.23 | 0.82 |
| DP Fearful | 66.58 | 399.50 | 60.71 | 6981.50 | 311.5 | −0.40 | 0.69 |
| Face bias | 50.25 | 301.50 | 61.56 | 7079.50 | 280.50 | −0.77 | 0.44 |
| Fear bias | 65.08 | 390.50 | 60.79 | 6990.50 | 320.50 | −0.29 | 0.77 |

DP – disengagement probability.

## Discussion

Results of this study show a preliminary suggestion that paternal sexual abuse in childhood might predict offspring's attentional disengagement patterns when viewing emotional facial expressions early in life. First, we found that paternal sexual abuse in childhood was negatively associated with face bias in their children. Second, when we explored the potential sex-specific effects, we found that daughters of fathers sexually abused during childhood (vs. daughters of sexually non-abused fathers) presented higher disengagement from neutral faces. The age-typical disengagement probability from fearful faces led to lower face bias and higher fear bias in their attention patterns. To the best of our knowledge, no previous study has explored the effects of paternal childhood adversities on emotional processing early in life in human offspring.

In our study, paternal sexual abuse in childhood was related to lower face bias in offspring. The experience of sexual abuse in early-life may shape later-life parenting and fatherhood involvement in raising offspring. Fathers who have been sexually abused in childhood have been shown to report higher levels of overprotective behaviors while performing parental activities [45]. It could be then speculated that offspring developing in conditions of potentially constant threat-apprehensiveness by the parent might themselves present less reactivity to non-fearful facial expressions. However, on the other hand, higher face bias (e.g., the ability to potentially discriminate between, e.g., friendly and non-friendly faces) in such children could also be beneficial for their capability to assess the surrounding environment and for their survival [6]. It has also been shown that sexually abused fathers present signs of hypervigilance concerning the offspring's safety [45]. This fact could also potentially contribute to the observed lower face bias in children and their faster distraction toward salient peripheral stimuli in the face-distractor paradigm. Another plausible reason for lower face bias in children fathered by sexually abused individuals in our study group is the composition of it, leveraged rather toward families with relatively high levels of education and sufficient socio-economic status. Such hypothetically "secure" conditions of development could also contribute to potentially lower attentional reactivity toward emotional stimuli.

Our results also show that at eight months, at the time of high fear preference in infant attention disengagement patterns, daughters of sexually abused fathers disengage more frequently from non-fearful facial expressions as compared to daughters of fathers who were not sexually abused. However, they are prone to engage in fear signals similarly to other infants. This pattern of attention leads to heightened salience and preference for fear in attentional processing.

The observed results may be considered as a continuation of our previous work in this cohort. We have observed an association between paternal early life adversities and offspring's brain structures, namely faster white matter maturation in the corpus callosum, larger grey matter volume, and higher amygdala volume [23–25]. We have also found that at 8 months of age, infants present a positive association between left amygdala volume measured at birth and attentional disengagement from fearful faces [26]. Our current results partially lie in accordance with the previous outcomes suggesting

a potentially "more advanced" brain development in daughters of fathers exposed to childhood maltreatment, as attention to faces is considered as a hallmark of neurocognitive development in infants. Yet, research in this area is currently scarce and this hypothesis requires further investigation.

A presumably "advanced brain" in daughters of fathers exposed to sexual abuse in childhood could be viewed from an evolutionary perspective. Higher bias towards fearful faces in infant daughters could appear as an intergenerationally inherited signal, leading to the ability to evaluate a potential source of threat [29]. Such intensified attention to fearful facial expressions could be considered an adaptive feature in order to increase the probability of survival. It has previously been related to more secure mother-infant attachment at 14 months [46]. Being more alert to a probable danger may be beneficial and desired, especially in females, as they bear highly aggravating costs of reproduction and provide primary care for the offspring. However, the question of whether such a potential threat-alerted phenotype is persistent into adulthood remains unanswered. Also, fathers who experienced childhood sexual abuse may present a more overprotective behavior towards the newborn, which might result in higher attachment of a child and thus higher fear bias. Potentially, this kind of paternal behavior could have a stronger effect in girls who may present more intensified reactivity to postnatal signals and recognize emotions better than boys [47]. As it would be highly informative to explore the potential effect of paternal caregiving behavior on emotional processing in the offspring in our study sample, yet no such data are available for this cohort.

One of the plausible molecular mechanisms that might stand behind the currently observed effects is epigenetic alterations to the paternal sperm. A few previous studies have reported evidence of epigenetic modifications, namely DNA methylation (DNAm) and small non-coding RNAs expressions (i.e., micoRNAs) in the sperm of human males exposed to childhood maltreatment [48,49]. Also, our recent study in the same FinnBrain cohort showed that paternal childhood adversities were linked to the altered expression of small non-coding RNA and DNAm in the sperm, including markers involved in brain neurodevelopment [25]. Going further, as studies on germline epigenetic inheritance in humans are still scarce, comparing to well-studied animal models [50], a recent epigenome-wide association study pioneered in showing that higher paternal childhood adversities are linked to the methylation levels at eight CpGs, the same CpGs that were later associated with attentional problems in children at 3 years of age [51]. What is also important to emphasize, from the viewpoint of our sex-specific results, is that epigenetic regulations may lead to sexually dimorphic effects and impact the brain function and structure of males and females differentially, e.g., by controlling the sex-hormone fluctuations [30]. However, the driving force and the exact role of the paternally inherited epigenetic alterations in the offspring, including their potential role in the sex-specific brain neurodevelopment, is still under debate and yet to be unveiled [52].

As we observed a significant association between paternal anxiety level at 6 months postpartum and offspring's face bias, we cannot exclude the possibility that parental post-natal interactions with a child could contribute to the observed results, despite the fact that eye-tracking data were collected relatively close to the birth. We performed a mediation analysis to explore if paternal postnatal anxiety levels could act as a mediator of the observed association between paternal ACEs and offspring's face bias, as well as between paternal sexual abuse during childhood and offspring's face bias. This analysis was exploratory by nature and was not originally hypothesized. We indeed revealed an indirect association between paternal TADS score (independent variable) and offspring's face bias (dependent variable) mediated by paternal anxiety level at 6 months postpartum (p = 0.031 for indirect pathway) (S6 Fig in S1 File, Electronic Supplementary Materials). However, we did not observe a significant indirect association between paternal sexual abuse during childhood and offspring's face bias mediated through paternal anxiety level at 6 months postpartum (p = 0.29 for indirect pathway) (S7 Fig in S1 File).

Another fact worth mentioning is that we were able to observe the effect of paternal childhood sexual abuse on emotional processing in the offspring despite the small sample size of sexually abused fathers (n = 12). It can be suggested that this type of adversity might be exceedingly stressful for the exposed individuals when compared to other types of adversities. Therefore, the effects of sexual abuse were detectable even in a limited number of affected fathers. Also, the potential uniqueness of sexual abuse in shaping the variation in several psychological and biological outcomes was recently pointed out [53].

The strength of the study is the use of innovative methodology to assess the attentional engagement to emotional facial expressions in children, namely eye-tracking. This procedure was conducted as early as possible in the post-natal development (at 8 months of age) when both face and fear biases are typically well detectable [8]. However, some limitations of the study should be considered. The sample size of fathers who were sexually abused (and therefore the number of sons and daughters born to the fathers who were exposed to sexual abuse, n = 12; 5% of the total sample) is small. This is also related to another limitation, which is the risk of underreporting or selective reporting, especially in men who are more prone to not disclose the fact of being a victim of sexual abuse than women [54] or to under-report the family history of addiction or mental illness, which might also be related to both ACEs and the offspring emotional processing. What is also worth mentioning is that the TADS questionnaire [35], used in the current study, allows for assessing solely the episodes of sexual abuse involving contact with the abuser, yet the non-contact sexual assaults might also be considered as an important element in the definition of sexual abuse. Also, we do not possess information about the accumulation of the adverse events (severity and frequency of sexual abuse episodes) in this sample. Another limitation could be a potential recall bias, as we assessed childhood adversities among adults. However, recall bias is a common issue in studies where events from many years ago are being reported. Others have previously demonstrated that a recall bias regarding childhood adversities seems not to compromise the validity of results in studies based on younger [55] and older adults [56]. Also, perhaps a multi-source assessment of childhood adversities involving information not only from an individual but also other sources (caregivers, school, medical records) could be a potentially more informative proxy of childhood events [14]. Lastly, we did not collect any additional information on how easily faces were encountered by children or any other information on how children were processing faces (apart from the eye-tracking procedure).

Future studies in this area are needed, especially those involving a larger sample size of the exposed individuals and more precise childhood adversity assessment in fathers. Additionally, longitudinal studies examining whether the observed effects persist into children's later life, potentially impacting their emotional processing and risk of diseases, would be highly informative.

To the best of our knowledge, this is a novel yet preliminary study suggesting the potential intergenerational effect of paternal childhood adversities on offspring's attentional disengagement from faces early in life. Further exploration of this topic will allow us to evaluate if future interventions, focused on eliminating the effects of childhood adversities, should consider the potential intergenerational continuity.

## Supporting information

**S1 File.   S1 Fig.** The distribution of paternal emotional neglect score. **S2 Fig.** The distribution of paternal emotional abuse score. **S3 Fig.** The distribution of paternal physical neglect score. **S4 Fig.** The distribution of paternal physical abuse score. **S5 Fig.** The distribution of paternal sexual abuse score. **S6 Fig.** Relationship between the paternal total TADS score and the offspring's face bias with paternal anxiety level at 6 months postpartum as a mediator. **S7 Fig.** Relationship between the paternal sexual abuse score and the offspring's face bias with paternal anxiety level at 6 months postpartum as a mediator.
(ZIP)

## Acknowledgments

We would like to greatly thank all the FinnBrain Families for devoting their time to our research.

## Author contributions

**Conceptualization:** Magdalena Klimek, Hasse Karlsson, Linnea Karlsson, Eeva-Leena Kataja.

**Formal analysis:** Magdalena Klimek, Hasse Karlsson, Eeva-Leena Kataja.

**Funding acquisition:** Hasse Karlsson, Linnea Karlsson, Riikka Korja, Saara Nolvi.

**Investigation:** Magdalena Klimek, Hasse Karlsson, Eeva-Leena Kataja.

**Methodology:** Riikka Korja, Saara Nolvi, Tuomo Häikiö, Jetro J. Tuulari, Eeva-Leena Kataja.

**Project administration:** Hasse Karlsson, Linnea Karlsson.

**Supervision:** Hasse Karlsson, Eeva-Leena Kataja.

**Visualization:** Eeva-Leena Kataja.

**Writing – original draft:** Magdalena Klimek, Eeva-Leena Kataja.

**Writing – review & editing:** Hasse Karlsson, Linnea Karlsson, Riikka Korja, Saara Nolvi, Tuomo Häikiö, Jetro J. Tuulari.

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
