## [Decision Letter · Decision Letter 0]

PONE-D-24-53637Paternal adverse childhood experiences and offspring’s attentional disengagement from faces at 8 months – results from the FinnBrain Birth Cohort StudyPLOS ONE

Dear Dr. Klimek,

Thank you for submitting your manuscript to PLOS ONE. After careful consideration, we feel that it has merit but does not fully meet PLOS ONE’s publication criteria as it currently stands. Therefore, we invite you to submit a revised version of the manuscript that addresses the points raised during the review process.

We look forward to receiving your revised manuscript.

Kind regards,

Maria Christine Magnus, PhD

Academic Editor

PLOS ONE

Journal Requirements:

3. You have indicated that data is available from [linnea.karlsson@utu.fi].  Please can we ask you to provide us with a general contact email address for the data requests, so readers can request access in perpetuity. If a general email is not available please provide a link to a website where readers can obtain access to data.

Additional Editor Comments (if provided):

I would like to apologize for the delay in the processing of the manuscript. This is due to difficulties identifying reviewers. The two reviewers have some relevant comments and suggestions for improvements. Please carefully adress these comments . In particular, the second reviewer has several concerns regarding the framework of the analysis from a causal inference framework.

Reviewers' comments:

Reviewer's Responses to Questions

**Comments to the Author**

1. Is the manuscript technically sound, and do the data support the conclusions?

Reviewer #1: Yes

Reviewer #2: Partly

2. Has the statistical analysis been performed appropriately and rigorously? 

Reviewer #1: Yes

Reviewer #2: No

3. Have the authors made all data underlying the findings in their manuscript fully available?

Reviewer #1: No

Reviewer #2: No

4. Is the manuscript presented in an intelligible fashion and written in standard English?

Reviewer #1: No

Reviewer #2: Yes

5. Review Comments to the Author

Reviewer #1: The study by Klimek and colleagues is interesting and touches a very vital topic Paternal ACE on intergenerational effects on attention at very early stage of neurodevelopment. While Brain carries the signature of ACE from early childhood because ACE has the maximum effect (Please see Kashyap et al, Psychological Medicine) on the brain (compared to all other factors), but whether the effects are carried out to next generation is definitely interesting and required. The methodology is also simple and easily understandable to readers. I have few issues that can be handled easily.

2.I believe the introduction can be shortened- Keeping the readers focus to the work done and shortening the Genetic-neuroimaging parts

3.Is there anyway to check for similarity of faces between the presented stimulus and faces encountered easily- (possibly done by any questionaires to parents)- I understand that the manuscript is from a concluded experiment, but still (else may be a limitation)

4.Please define each model - (in equation form) above its table. Also provide more parametric values from the tests for model validity.

5.I would like to see a more focussed discussion pertaining to the outputs of the study- Most of the studies they have cited can go in future direction including genetics , neuroimaging etc.

Minor-

Spelling mistakes exist

Shorten the introduction and discussion

Reviewer #2: Thank you for inviting me to review this manuscript. The authors attempt to relate paternal ACEs to offspring’s attentional disengagement from faces at 8 months. Below are my comments for the authors:

• How were the 1,953 fathers from the FinnBrain Birth Cohort Study selected? For example, was this a random or convenience sample? The same question applies to the 239 children. Including a flowchart would be helpful.

• Table 1 is somewhat limited. Do you have any additional information about your sample? For example: whether fathers are still in a relationship, whether children were born prematurely, any relevant medical history of the children, parents’ education levels, parents’ psychiatric history, childcare arrangements, family history, etc.

• On page 13, you state: “There was no difference between the studied group and the total sample, etc.” Does this comparison refer to the 239 fathers versus the 1,953 fathers? What about comparisons with all fathers from the FinnBrain Birth Cohort Study?

• On page 14: “If a statistically significant correlation was observed, then the covariate was included in the final statistical model.” This approach is not ideal. Covariate selection should be based on a conceptual framework, independent of statistical correlation.

• Depression and anxiety at 6 months postpartum may be on the causal path between paternal ACE and attention to faces in offspring. Including these mediators as covariates in your model could bias your causal estimate.

• Along these lines, your Table 3 is subject to the so-called “Table 2 fallacy” (https://pubmed.ncbi.nlm.nih.gov/23371353/), especially regarding paternal anxiety and maternal EPDS being on the causal path.

• Still related to your causal framework, your model may not adequately account for confounders. For example, paternal ACEs may be influenced by factors that also affect children’s attentional disengagement, such as family history of addiction or mental illness. Ideally, you would conduct additional analyses to account for these factors. If these data were not collected, this limitation should be acknowledged.

6. PLOS authors have the option to publish the peer review history of their article (what does this mean? ). If published, this will include your full peer review and any attached files.

**Do you want your identity to be public for this peer review?** For information about this choice, including consent withdrawal, please see our Privacy Policy .

Reviewer #1: **Yes: ** Rajan Kashyap

Reviewer #2: No

---

## [Author Response · Author response to Decision Letter 1]

26 May 2025

We sincerely thank the Reviewers for their valuable input, which has greatly improved the clarity and rigor of our manuscript. We have carefully considered each suggestion and revised the manuscript accordingly, whenever it was possible. We trust that these revisions address the Reviewers' concerns. Below, we provide a point-by-point response to all comments. All changes made to the manuscript are highlighted using tracked changes.

---

## [Editor Report · Decision Letter 1]

Paternal adverse childhood experiences and offspring’s attentional disengagement from faces at 8 months – results from the FinnBrain Birth Cohort Study

PONE-D-24-53637R1

Dear Dr. Klimek,

We’re pleased to inform you that your manuscript has been judged scientifically suitable for publication and will be formally accepted for publication once it meets all outstanding technical requirements.

Kind regards,

Maria Christine Magnus, PhD

Academic Editor

PLOS ONE

Additional Editor Comments (optional):

Thank you for your responding in detail to the comments raised by the reviewers. Overall, I think your manuscript has approved considerably.
---

## [Editor Report · Acceptance letter]

PONE-D-24-53637R1

PLOS ONE

Dear Dr. Klimek,

I'm pleased to inform you that your manuscript has been deemed suitable for publication in PLOS ONE. Congratulations! Your manuscript is now being handed over to our production team.

Kind regards,

on behalf of

Dr. Maria Christine Magnus

Academic Editor

PLOS ONE